# Biofilm Formation of *Clostridioides difficile*, Toxin Production and Alternatives to Conventional Antibiotics in the Treatment of CDI

**DOI:** 10.3390/microorganisms11092161

**Published:** 2023-08-26

**Authors:** Leon M. T. Dicks

**Affiliations:** Department of Microbiology, Stellenbosch University, Stellenbosch 7600, South Africa; lmtd@sun.ac.za

**Keywords:** *Clostridioides difficile*, colonization, toxin production, pathogenicity of toxins, antibiotics, fidaxomicin, sequestering or inactivating toxin production, quorum quenching, immunization, bacteriotherapy

## Abstract

*Clostridioides difficile* is considered a nosocomial pathogen that flares up in patients exposed to antibiotic treatment. However, four out of ten patients diagnosed with *C. difficile* infection (CDI) acquired the infection from non-hospitalized individuals, many of whom have not been treated with antibiotics. Treatment of recurrent CDI (rCDI) with antibiotics, especially vancomycin (VAN) and metronidazole (MNZ), increases the risk of experiencing a relapse by as much as 70%. Fidaxomicin, on the other hand, proved more effective than VAN and MNZ by preventing the initial transcription of RNA toxin genes. Alternative forms of treatment include quorum quenching (QQ) that blocks toxin synthesis, binding of small anion molecules such as tolevamer to toxins, monoclonal antibodies, such as bezlotoxumab and actoxumab, bacteriophage therapy, probiotics, and fecal microbial transplants (FMTs). This review summarizes factors that affect the colonization of *C. difficile* and the pathogenicity of toxins TcdA and TcdB. The different approaches experimented with in the destruction of *C. difficile* and treatment of CDI are evaluated.

## 1. Introduction

*Clostridioides difficile* is a Gram-positive, sporulating, rod-shaped cell, and obligatory anaerobic [1]. Although *C. difficile* is considered a nosocomial pathogen that flairs up in patients exposed to antibiotic treatment [2,3], four out of ten patients diagnosed with *Clostridioidis difficile* infection (CDI) acquired the infection from non-hospitalized individuals [4], many of whom have not been treated with antibiotics [5]. Contracting *C. difficile* may also be through contact with infected animals, including reptiles and birds [6,7]. One to three percent of adults are asymptomatic carriers of *C. difficile* [8].

Approximately half a million people in the USA are hospitalized with CDI annually, and 5 to 6% die within the first month of diagnosis [9]. According to Mada and Alam [9], antibiotic use remains the leading risk factor for *C. difficile* infection. Of these, penicillins, cephalosporins, fluoroquinolones, and clindamycin have been implicated as the most possible cause [9]. Other risk factors associated with CDI include advanced age, chemotherapy, use of proton pump inhibitors, chronic renal disease, chronic liver disease, and malnutrition [9]. Based on the latest report from the Center for Disease Control (CDC), recurrent *C. difficile* infection (RCDI) was reported in 12.0% of the 4301 cases studied, with a sharp increase in 2020 during the onset of the COVID-19 pandemic [10]. An earlier report by Miranda-Katz et al. [11] stated that only 17 to 24 in 100,000 children develop CDI, which is eightfold lower than that reported in adults over the age of 65. The resistance of infants and young children to CDI may be ascribed to the immunoglobulin in breast milk that inhibits the binding of TcdA to intestinal receptors, along with the lack of intestinal receptors in newborns that recognize the toxin [12,13]. With aging, changes in diet and the secretion of bile acids render intestinal cells more susceptible to *C. difficile.* Drastic changes in the gut microbiota, as observed with prolonged antibiotic treatment, prevent the conversion of primary bile acids to secondary bile acids. This favors *C. difficile* colonization [14,15]. Biofilm formation protects cells from oxygen and antibiotics, including metronidazole (MNZ) and vancomycin (VAN), which are commonly used to treat CDI [16,17].

Treatment of recurrent CDI (rCDI) with antibiotics, especially vancomycin (VAN) and metronidazole (MNZ), increases the risk of experiencing a relapse by as much as 70% [16,17]. Strains becoming resistant to both antibiotics are on the increase. Fidaxomicin, which prevents the initial transcription of RNA toxin genes, proved more effective than VAN and MNZ. The efficacy of antibiotics is, however, hampered by their poor ability to penetrate biofilms. More research is required on alternatives to antibiotics, such as non-antimicrobial agents sequestering or inactivating toxin production, quenching of genes (quorum quenching, QQ), immunization, and bacteriotherapy, including fecal microbial transplants (FMTs).

The first part of this paper summarizes the factors affecting the colonization of *C. difficile* in the gastrointestinal tract (GIT), toxin production, and the pathogenicity of toxins TcdA and TcdB. The advantages and disadvantages of antibiotics are discussed, and results obtained with fidaxomicin are compared to treatment with MNZ and VAN. Alternatives to antibiotics, such as non-antimicrobial agents sequestering or inactivating toxin production, quenching of genes (quorum quenching, QQ), immunization, bacteriotherapy, and microbiome replacement therapies are reviewed.

## 2. Colonization of *C. difficile* to Human Intestinal Cells

CDI is contracted through the ingestion of endospores [18]. Germination of spores is controlled by the concentration and type of primary bile salts in the upper part of the GIT [19,20]. Chenodeoxycholate (CDCA) represses spore germination, whereas cholate (CA) induces germination [20]. Most of the primary bile acids (95%), conjugated with taurine and glycine or unconjugated, are absorbed in the terminal ileum and through the hepatic system [19,21]. Primary bile acids that reach the large intestine are converted by gut microbiota into secondary bile acids, for example, ω-muricholate (ωMCA), hyodeoxycholate (HDCA), ursodeoxycholate (UDCA), lithocholate (LCA), and deoxycholate (DCA) [19,22].

Physiological conditions, such as an excess of fermentable carbohydrates or an increase in deoxycholate (DOC, Figure 1), stimulate *C. difficile* to form biofilms in the human GIT, which may lead to recurrent episodes of CDI [23,24]. Biofilm formation is also regulated by quorum sensing (QS) signals such as cyclic diguanosine monophosphate (c-di-GMP; Figure 1).

Increased production of c-di-GMP represses motility and stimulates biofilm formation [25]. The synthesis of c-di-GMP is controlled by the protein domain GGDEF, which is widely present in free-living bacteria [26,27,28,29]. Increased c-di-GMP levels reduce the expression of *tcdA, tcdB,* and *tcdR* [25,26,27,28,29,30]. The *tcdR* gene encodes an alternative sigma factor SigD (FliA; σ^28^) that activates the expression of *tcdA* and *tcdB* in response to c-di-GMP [25,30]. Degradation of c-di-GMP is controlled by the protein domain EAL (Figure 1) [26,27,28,29].

Changes in c-di-GMP levels influence the response of riboswitches, which in turn control the expression of flagellar genes. Biological functions have been assigned to 11 of the 16 riboswitches described for *C*. *difficile* [31,32]. Seven of the riboswitches, classified as class I, behave in an “off” position in the presence of high levels of c-di-GMP, that is, they terminate gene transcription. The remaining four functional riboswitches, defined as class II, react as “on” switches and trigger gene expressions. Elevated levels of c-di-GMP prevent the transcription of flagellar genes [33,34] and result in biofilm formation (Figure 1, left panel). Strains with mutations in flagellar genes *fliC* and *fliD* produced higher levels of TcdA and TcdB [35]. A decrease in the transcription of *sigD*, located on operon *flgB*, represses the expression of genes encoding the synthesis of chemotaxis proteins, cell wall proteins (e.g., collagen-binding protein CbpA), and putative membrane transport proteins [36]. Mutations in *sigD* and flagellar genes *fliF*, *fliG, and fliM* resulted in loss of motility and a significant decrease in the expression of toxin genes [30].

Binding of c-di-GMP to Cdi-2-4, one of the four class II c-di-GMP riboswitches located directly upstream of the type IV pili (T4P) primary locus, upregulates the transcription of TFP (type IV pilus) genes and stimulates the aggregation of *C. difficile* cells [37]. In the absence of c-di-GMP, Cdi-2-4 induces transcription termination and prevents the expression of TFP genes [38]. A 2.54-fold reduction in the expression of *fliC* in the hypervirulent *C*. *difficile* strain R20291 stimulated flagellin production and biofilm formation on glass beads after 7 days [39,40]. Downregulation of other flagellar biosynthesis genes such as *flhA*, *flbD*, *flgE,* and *flgD* also resulted in biofilm formation [41].

Our understanding of biofilm formation by *C*. *difficile* is far from complete, as stimuli for the aggregation of hypervirulent strains (e.g., strains 630 and R20291) differ [42]. Valiente et al. [43] reported an increase in cell hydrophobicity for strain R20291 that lacked flagellar post-transcriptional modification. Only two of the five mutants studied had reduced motility; however, all five mutants showed an increase in biofilm formation. This led the authors to conclude that biofilm formation by *C. difficile* is not influenced by motility but by hydrophobicity due to the presence of glycan on the flagella. Dapa et al. [44] suggested that flagella play an important role in the late stages of biofilm formation, as shown with the mutant R20291 *fliC* ClosTron. Since the pilin *pilA1* gene (CD3513) in *C. difficile* is regulated by c-di-GMP acting on the upstream riboswitch Cdi-2-4 [32,45], pili may be required for initial adhesion to epithelial cells and initiate biofilm formation [34]. Pili does, however, not promote late-stage biofilm formation [39].

The importance of cell structure in biofilm formation cannot be ignored. Poquet et al. [41] showed that cells in biofilms have upregulated phospholipid metabolism, active acyl carrier proteins, and increased fatty acid synthesis. Increased production of fatty acids was also reported for cells of *Bacillus subtilis* in biofilms [46]. Mutants of *C. difficile* with a deficient *lcpB* gene and inability to deposit PSII teichoic acids at the cell surface [41] were elongated, larger in diameter, formed abnormal septa, and grew slower [47,48,49]. Cell wall-binding protein Cwp11 (CD2795), cell surface protein Cwp10 (CD2796), and calcium-binding adhesion protein (CD2797) are regulated by c-di-GMP [32]. Cwp11 is released in the “secretome” during biofilm formation [41,50].

A mutation in the *prkC* gene of *C. difficile* 630Δ*erm* resulted in increased biofilm formation after 24 h, but only in the presence of bile salt DOC [51]. The function of *prkC* in *C*. *difficile* remains unknown [51]. Mutation of the *dnaK* gene of strain 630Δ*erm* resulted in the disruption of DnaK synthesis and thus protein folding but also led to a significant increase in biofilm formation and cell elongation [52]. Similar results were recorded when *lexA*, which encodes the transcriptional repressor LexA in *C. difficile* R20291, was disrupted [47,53]. Other genes attributed to *C. difficile* biofilm formation are *spo0A* [23,24,54], quorum sensing regulator *luxS* [44,45], and germination receptor *sleC* [54]. Inactivation of the chaperones *dnaK* and *hfq* changes the cells to become temperature-sensitive and increases biofilm formation [52,55].

Iron plays a key role in the growth of pathogenic bacteria, including *C. difficile* [56]. Ferrous iron is required for *C. difficile* to colonize the large bowel [57]. *C. difficile* regulates iron transport with three membrane-bound ferrous iron transporters (FeoBs), of which FeoB1 is produced in the highest quantity under iron-limiting conditions [58]. Although iron stimulates the growth of *C. difficile* and renders the species more resistant to MNZ [59], it is not known whether an increase in FeoB1 leads to elevated levels of TcdA and TcdB.

Extracellular DNA (eDNA) is a major component of *C*. *difficile* biofilms [44,60]. Hypervirulent *C. difficile* 027 strains are rich in prophages and mobile genetic elements [61]. DNA released from lysed cells may support biofilm formation, as observed for *Staphylococcus aureus* and *Pseudomonas aeruginosa.* In both these species, cell lysis in biofilms is controlled by signals regulating quorum sensing [62,63,64]. In *C*. *difficile* biofilms, S-ribosylhomocysteinase (LuxS) induces prophages, which likely contribute to biofilm formation [55]. In LuxS mutants, on the other hand, downregulated prophage loci are conserved among *C*. *difficile* strains, specifically region 2 encoding a phiC2-like phi-027 phage [40,61,65].

In a complex system, such as the human GIT, metabolites and enzymes produced by bacteria have a profound influence on the microbial population. Bile salt hydrolase (BSH), for instance, produced by *Bacteroides ovatus*, inhibited the growth of *C. difficile* [66]. *Bacteroides fragilis* inhibited the growth of wild-type strains of *C. difficile* in biofilms when cultured together [55]. Elevated levels of succinate produced by *B*. *fragilis* increased the regulation of succinate metabolism by *C*. *difficile* [41,67]. With the upregulation of *sucD* (succinate-CoA ligase [ADP-forming] subunit alpha), increased expressions of *accB* (biotin carboxyl carrier protein of acetyl-CoA carboxylase), *abfH* (4-hydroxybutyrate dehydrogenase), *abfT* (4-hydroxybutyrate CoA-transferase), *abfD* (4-hydroxybutyryl-CoA dehydratase/vinylacetyl-CoA-delta-isomerase), and *cat1* (catalase-1) were noted [55]. Other genes of *C. difficile* were downregulated, e.g., *bcd2* and *idhA,* encoding butyryl-CoA dehydrogenase and (r)-2-hydroxyisocaproate dehydrogenase, respectively [55].

When cells of *C. difficile* deficient in *luxS* were co-cultured with *B*. *fragilis*, biofilm formation by the mutant was much weaker than when the same experiment was performed with the wild-type strain of *C. difficile* [55]. This indicated that AI2/LuxS is involved in facilitating *B*. *fragilis*-induced inhibition of *C*. *difficile*. Poquet et al. [41] also reported the downregulation of genes involved in carbohydrate metabolism. Both studies have shown that changes in carbohydrate metabolism favor the growth of *B*. *fragilis* at the expense of *C*. *difficile*. Thus, repression of carbohydrate metabolism plays a major role in biofilm formation. However, the signaling molecules orchestrating the downregulation of genes involved in key metabolic pathways are unknown. Planktonic cells of *B. fragilis* and *C. difficile* have no effect on each other’s growth [55], suggesting that the cells must be in close contact with each other. Biofilm formation by *C. difficile* is a multifaceted and complex process. For more information on biofilms and hypervirulence, refer to Taggart et al. [42].

## 3. Intra- and Inter-Cellular Communication

Two communication or quorum sensing (QS) systems have been identified in *C. difficile*, i.e., an inter-species LuxSCD system (top section of Figure 2) and an intra-species accessory gene regulator (Agr) system (bottom section of Figure 2). Genes encoding homologues of luxS have also been detected in *C. difficile* [68,69,70,71,72,73]. The *luxS* gene encodes AI-2 synthase (LuxS). Downstream of *luxS* are *orfX* and *metH* [74]. The function of *orfX* is unknown. The *metH* gene encodes 5-methyltetrahydrofolate-homocysteine methyltransferase. AI-2 molecules induce the transcription of the toxin genes *tcdA*, *tcdB,* and *tcdE* during the late exponential growth phase and modulate biofilm formation. Products of *rolA* and *rolB* upstream of the *luxS* gene act as negative regulators of AI-2 [74].

Studies conducted on a mutant of *C. difficile* with defective luxS have shown that biofilm formation could be restored by supplementing the growth medium with 4,5-dihydroxy-2, 3-pentanedione (DPD), the precursor of AI-2 [55]. As little as 100 nM of DPD was sufficient to restore biofilm formation. This indicates that AI-2 may be involved in signaling among *C. difficile* biofilm cells. Interestingly, no significant differences in biofilm formation or luxS expression were observed in *C. difficile* strains isolated from patients with recurrent and non-recurrent CDIs [75]. This suggests that other unknown regulatory systems or QS signals are involved in the colonization of *C. difficile*. Strains from recurrent CDI sporulated more [75]. This observation needs to be studied in more depth, as the genes involved in sporulation may influence colonization and biofilm formation.

The cell surface receptors for *C*. *difficile* have not yet been identified, and the LuxS/AI-2 mechanism, especially within biofilm communities, is unknown. A mutant defective in LuxS (strain R20291 *luxS* ClosTron mutant) did not produce AI-2 and could not form a biofilm in vitro [55]. RNA sequencing of genes expressed by R20291 *luxS* mutant cells in biofilms showed an increase in the expression of CDR20291_2554 (*crr*), a phosphotransferase (PTS) glucose-specific transporter subunit IIA; CDR20291_2927, a cellobiose phosphate-degrading protein; and CDR20291_2930 (*treA*), a trehalose-6-phosphate hydrolase [55]. An increase in the degradation of trehalose and the functioning of PTS (genes encoding these are on the same operon) provides *C*. *difficile* a competitive advantage over gut microbiota. Trehalose acts as an osmoprotectant [76] and prevents protein (and thus toxin) re-conformation during dehydration. Thus, it is possible that trehalose plays an important role in the formation of luxS-mediated *C*. *difficile* biofilms, similar to what has been reported for *Candida albicans* [77].

## 4. Toxin Production

Toxin TcdA, classified as an enterotoxin (2710 amino acids; 308 kDa), causes accumulation of fluid in the ileum [78]. Toxin TcdB, a cytotoxin consisting of 2 366 amino acids (270 kDa), is 100- to 1000-fold more potent than TcdA [79]. Genes encoding the two toxins, *tcdA* and *tcdB*, are located on a 19.6 kb pathogenicity locus, PaLoc, together with regulatory genes *tcdR, tcdC,* and the toxin secretion gene *tcdE*, as shown in Figure 3 [80,81]. The *tcdR* gene encodes a sigma factor that controls the transcription of the toxin promoters and its own promoter [82,83]. The *tcdC* gene encodes an anti-sigma factor, TcdC, that deregulates toxin synthesis [84,85]. Neither the deletion of *tcdC* nor the altering of *tcdC* frameshift mutations influenced toxin synthesis [86,87]. Regulation of toxin synthesis is thus more complex and includes the involvement of other key regulatory elements. Darkoh et al. [88] have shown that toxin synthesis is regulated by an Agr QS system. *sigD* upregulates *tcdR* [25,30]. The expression of *sigD* is repressed by elevated levels of c-di-GMP, which in turn downregulate the expression of *fliC* and toxin production [25].

Pathogenic strains of *C. difficile*, such as R20291, contain two accessory gene regulator (Agr) loci, *Agr1* and *Agr2* (Figure 2). The *Agr1* locus contains *agrB1* (encoding a transmembrane protein AgrB1) and *agrD1,* encoding a prepeptide AgrD1, which produces an auto-inducer peptide (AIP) called thiolactone or “TI signal.” The latter acts as a signaling molecule to induce *C*. *difficile* toxin production once it reaches a sufficient concentration. Treatment of the prepeptide with hydroxylamine disrupted the thioester bonds [89,90,91] and resulted in the loss of activity, indicating that TI contains cysteine residues. According to Darkoh et al. [92], TI is less than 1000 Da in size and is constitutively produced in the GIT by hypervirulent strains of *C. difficile*. Deletion of the *Agr1* locus resulted in the loss of toxin production in both the hypervirulent strain R20291 and non-hypervirulent strain 630 [93]. No significant mRNA transcripts from genes *tcdA* and *tcdB* were detected in any of the two strains. However, gene transcripts were detected in the R20291 *Agr2* mutant and the wild-type strains R20291 and 630. No toxin activity was detected in cell-free supernatants collected from *Agr1* mutants when tested in fibroblast cells. Based on these findings, the *Agr1* locus appears to play a central role in toxin production. This finding was confirmed in vivo. The mutant strain of *C. difficile* (R20291 *Agr1* mutant) colonized mice but did not develop CDI. The *Agr2* locus contains quorum signal generation (*agrB2D2*) and response (*agrC2A2*) genes. Non-hypervirulent strains such as 630 contain only the *Agr1* locus. At a certain threshold, TI peptides interact with the two-component AgrC2 histidine kinase, catalyzing the phosphorylation and dimerization of the response regulator AgrA2. The latter induces toxin production, either directly or indirectly. By altering the sensitivity of AgrA2 to TI, toxin production may be prevented. Whether this approach to CDI treatment is feasible remains uncertain. Strains of *S. aureus* lacking the *agr* gene (Δ*agr*) are more prone to cause chronic infections and bacteremia. Thus, the treatment of *S. aureus* infections using the QQ approach is not an option. Part of the *agr* QS locus in *S. aureus*, *agrDB* (*agr1*) is also present in *C*. *difficile* [40,93].

In hypervirulent *C*. *difficile* strains belonging to RT (ribotype)-017 and RT-027, QS is dependent on the expression of *agrACDB* in the *Agr2* operon [40,93,94]. This is similar to the *agr* QS locus *agrACDB* in *S*. *aureus*, which regulates virulence gene expression [94]. The *Agr2* QS system in *C. difficile* uses the cyclic AIP encoded by *agrD* and exported by the transmembrane protein encoded by *agrB* [94]. Strains with mutations in *AgrB1* and *AgrD1* lost the ability to transcribe *tcdA* and *tcdB*. The virulence of the *AgrB1D1* mutant was also reduced, as measured in a murine model of *C. difficile* infection [88,93]. Thus, the two-gene *Agr1* system is important for *C. difficile* pathogenesis, but its influence on virulence factor gene expression in the absence of a two-component system is unclear.

A significant increase in *agrD1* expression was noted in strains from recurrent CDI compared to non-recurrent CDI [75]. Strains with mutations in *agrB1* and *agrD1* of strains 630 and R20291 were deficient in both toxin A and B production [93]. A reduction in the expression of *tcdA* was observed in R20291 ClosTron mutants [94]. No differences in toxin expression (on RNA level) were detected between the R20291 *luxS* disruption mutant and the wild-type strain, suggesting that the LuxS QS system has little effect on toxin production in *C*. *difficile* [55]. An insertion mutation in *agrA* of strain R20291 also resulted in decreased expression of three genes encoding diguanylate cyclase (DGC) and phosphodiesterase (PDE) responsible for c-di-GMP production [94]. This implies that the Agr QS system is also involved in regulating the production of c-di-GMP [94]. Toxin production may be prevented by blocking AgrB1, eliminating the TI signal with analogs or antibodies, preventing the binding of AgrC2 to the TI signal, or preventing the phosphorylation or dimerization of AgrA2. Growth is not affected by the Agr system, which means that the strains may not become resistant to targeting the quorum signaling mechanism.

The Agr system in *C. difficile* 630 has multiple functions, as shown by Ahmed et al. [95]. Deletion of *agrB1* and *agrD1* by a Cas9 nickase system (CRISPR-Cas9n), or deletion of the entire locus, resulted in changes in gene expressions associated with sporulation. At the same time, the motility of *C. difficile* was reduced when these two genes were disrupted. Loss of AgrB1 resulted in the accumulation of AgrD1, which led to a 15-fold increase in the expression of *tcdR,* and a 20-fold and 5-fold increase in the expression of *tcdA* and *tcdB,* respectively. Deletion of *agrB1* and agrD1 or deletion of only *agrD1* did not significantly alter the expression of *tcdR* and tcdB but did result in a minor decrease in *tcdA* expression. In conclusion, the Agr1 system in *C. difficile* 630 performs multiple functions and not only AIP signaling. Agr1 influences sporulation efficiency, by requiring a combination of AgrB1 and AgrD1 [95]. Toxin expression is, however, only affected by the absence of AgrB1 and the intracellular accumulation of the AgrD1 peptide. Thus, Agr1 influences *C. difficile* activity via both AgrB1-dependent and AgrB1-independent mechanisms.

The production of TcdA and TcdB increases when cells in biofilms reach a certain threshold [25]. However, genes encoding the toxins are not triggered at the same rate and seem to be strain-related. The expression of *tcdB* increased significantly (2.83-fold) in biofilms of *C*. *difficile* R20291, in contrast to the expression of *tcdA* [39]. Toxins released into the colon are taken up by epithelial cells via receptor-mediated endocytosis. This causes mono-glucosylation of low-molecular-weight GTPases in the cytosol [79], resulting in the interruption of Rho GTPases, which leads to apoptosis, cell rounding, dysregulation of the actin cytoskeleton, and changes in cellular signaling [88]. These changes stimulate the release of several immunomodulatory mediators from epithelial cells, phagocytes, and mast cells, resulting in inflammation and the accumulation of neutrophils [88].

Biofilm cells of *C*. *difficile* 630Δ*erm* grown in a continuous-flow microfermentor have shown a 1.03-fold decrease in the expression of *tcdA* but no significant changes in the expression of *tcdB* [41]. Although results generated by Maldarelli et al. [39] and Poquet et al. [41] made use of different strains and the results are not directly comparable, toxin production is clearly much higher amongst cells in the biofilm. It is interesting to note that the expression of *tcdA* by *C*. *difficile* 630 in human fecal water decreased fourfold [89]. This was ascribed to a decrease in butyrate production [89]. Genes involved in the metabolism of pyruvate, such as *bcd2* and *idhA*, encoding butyryl-CoA dehydrogenase and (r)-2-hydroxyisocaproate dehydrogenase, respectively, were downregulated [55]. Such a shift in metabolism may stimulate the growth of *Bacteroides fragilis* and outcompete *C. difficile*. A decrease in toxin production coincided with a 300-fold increase in the expression of sporulation genes [89]. Concluded from these studies, there is no direct correlation between toxin production and endospore formation. This also suggests that pathogenicity, as far as toxin production is concerned, is highly dependent on biofilm formation. Tijerina-Rodrı’guez et al. [90] found no significant difference in the distribution of *C*. *difficile* ribotypes between recurrent and non-recurrent CDI cases. However, the authors reported significantly higher sporulation in recurrent CDI samples, especially increased expression of the sporulation genes *spo0A* and *sigH.* This implies that recurrent CDIs are more likely to be caused by an increase in the number of endospore-forming biofilm cells. It is, however, uncertain whether sporulation influences biofilm formation, as no significant differences in spore levels were recorded in 7-day-old biofilms of recurrent and non-recurrent CDIs [90]. This raises the question of whether sporulation plays a bigger role than biofilm formation in recurrent CDIs and requires more debate on the conditions required for toxin production.

## 5. Mode of Action of TcdA and TcdB

Both toxins, TcdA and TcdB, are composed of multimodular structures (a glucosyltransferase domain, GTD; cysteine protease domain, CPD; translocation domain, TD; and combined repetitive oligopeptide, CROP domain), as shown in Figure 4. The CROP domain at the C terminal is also referred to as the receptor-binding domain (RBD). Both toxins inactivate Rho- and Ras-GTPases in the host cytosol via mono-*O*-glucosylation (Figure 4). This leads to the disruption of tight junctions between intestinal cells, immune modulation [91], and inactivation of enzymes such as phospholipase D [92], protein and lipid kinases, and nicotinamide adenine dinucleotide-oxidase [93,94,95]. Inactivation of Rho proteins also leads to the induction of apoptosis [96], prevention of gene transcription, and the inhibition of phagocytosis [97], G1 cell cycle progression, microtubule dynamics, and vesicular transport pathways [93,94,95]. Notable symptoms caused by TcdA and TcdB are diarrhea, pseudomembranous colitis, and toxic megacolon [98,99].

## 6. Treatment of CDI with Antibiotics

Treatment of CDI with antibiotics is difficult because of biofilms that protect *C. difficile* [16,17]. MNZ is administered at 500 mg three times per day, whereas VAN is prescribed at 125 mg four times daily. In more complicated CDI cases (presence of hypotension, ileus, shock, or megacolon), oral VAN with or without intravenous MNZ is recommended [100]. VAN disrupts normal gut flora, whereas fidaxomicin, a narrow-spectrum macrocyclic antibiotic that inhibits the synthesis of RNA polymerase, causes less alteration of gut microbiota [101]. VAN and MNZ have been associated with the colonization of VAN-resistant enterococci (VRE) [102,103]. Recurrent infections occur in approximately 20% to 30% of patients [104,105], with higher CDI recurrence rates observed in patients who have experienced multiple episodes [106] and in subgroups of high-risk patients (oncology, renal impairment, concomitant antibiotics, increased age, and previous CDI episode) [104,107,108,109].

Clinical trials published in 2014 found MNZ inferior to VAN [110]. It is important to note that subinhibitory concentrations of MNZ stimulate *C. difficile* and increase biofilm formation [111,112]. It is also noteworthy that treatment of recurrent CDI (rCDI) with antibiotics increases the risk of experiencing a relapse by as much as 70% [18,113]. Failure of treatment with MNZ and VAN led to the treatment of CDI with fidaxomicin [114,115]. In 2011, the FDA approved fidaxomicin (previously known as OPT-80, PAR-101, Tiacumicin B and Difimicin) for the treatment of CDI in adults [116], and in 2020, for the treatment of CDI in children 6 months and older [117]. Since the approval of fidaxomicin, it has mainly been prescribed to treat multiple CDI recurrences [118,119]. In 2021, the Infectious Diseases Society of America (IDSA) and the Society for Healthcare Epidemiology of America (SHEA) updated CDI guidelines to encourage the use of fidaxomicin as first-line therapy in adults [120]. Moderate to severe cases of CDI are treated with fidaxomicin for 10 to 14 days [121]. The safety of fidaxomicin and its superiority to VAN was demonstrated in several clinical trials [105,122,123,124,125]. Patients that have been treated with VAN or fidaxomicin and experience their first relapse of CDI may be treated with the same antibiotics. Second and continuing relapses are treated with pulsed or tapered doses of vancomycin [126]. For more information on the resistance of *C. difficile* to antibiotics, the reader is referred to Dureja et al. [127].

Fidaxomicin is produced by the actinomycete *Dactylosporangium aurantiacum* subsp. *Hamdenesis.* It is an unsaturated macrocyclic lactone ring with a hepta-carbohydrate at position 12 and a 6-deoxy sugar at position 21 [100]. Isobutyryl ester at the fourth position is converted by an unknown esterase to produce metabolite OP-1118, which is also active against *C. difficile*. Fidaxomicin has a narrow spectrum of antibacterial activity and kills strains of *C. difficile* resistant to cephalosporins, fluoroquinolones, clindamycin, and rifamycin [128]. Isolates of *C. difficile* resistant to rifamycins or to other antimicrobial classes (cephalosporins, fluoroquinolones, and clindamycin) are not cross-resistant to fidaxomicin [129]. Artsimovitch et al. [115] were the first to show that fidaxomicin inhibits the binding of RNAP to the σ-specificity factor (Figure 5), thus preventing the opening and closing of the DNA:RNA clamp during the initiation of transcription. This means that holo RNAP is not formed. The addition of fidaxomicin after the formation of the transcription-competent open promoter complex did not inhibit transcription during the binding of the promoter to DNA. This clearly indicates that fidaxomicin blocks transcriptional initiation when template DNA strands separate and at the point before RNA synthesis starts. These transcription reaction experiments were performed in vitro in *E. coli* by adding fidaxomicin at specific time points during DNA translation. This mode of action distinguishes fidaxomicin from elongation inhibitors, such as streptolydigin, and transcription initiation inhibitors, such as myxopyronin and rifamycin [130,131]. Although rifamycins are also RNAP inhibitors, their efficiency and versatility are limited by the rapid increase in drug-resistant strains. This is because they act on the β-subunit, which is relatively dispensable [132,133].

Strains of *C. difficile* resistant to fidaxomicin had mutations in either the *rpo*B (Gln1074Lys or Val1143Phe) or *rpo*C (Asp237Tyr) genes close to the binding site of RNAP [130]. A Val1143Asp mutation resulted in delayed growth [134]. More than 92% of fidaxomicin, orally administered to adults at single doses of 200 and 300 mg, was recovered in feces, and approximately 0.59% in urine, indicating that it is minimally absorbed in the bloodstream. Fidaxomicin also inhibits the growth of methicillin-resistant *S. aureus* (MRSA), vancomycin-resistant *S. aureus* (VRSA), and *Mycobacterium tuberculosis*. However, the low solubility and low systemic bioavailability of fidaxomicin have precluded its use for the treatment of MRSA, VRSA, and tuberculosis.

## 7. Toxin Binding or Suppression

Targeting the Agr1 quorum signaling system, for example, blocking the synthesis of TI (the toxin-auto-inducing peptide) through AgrB1, sequestering the activity of TI with analogs or antibodies, preventing the binding of a histidine kinase (AgrC2) to TI, and preventing the phosphorylation or dimerization of the response regulator (AgrA2), may sequester toxin-induced damage and inflammation without drastic alterations of the gut microbiome. With a healthy gut microbiome and, hence, a healthy immune system, *C. difficile* should be naturally cleared from the GIT tract to decrease the risk of CDI relapse. This also lowers the chance of developing resistance, as in the case of antibiotic treatment.

Several steps of QS (Figure 2) may be blocked to prevent toxin production. QQ refers to (i) blocking AIP synthesis, (ii) interfering with QS by using analogs or antibodies, (iii) preventing the binding of AIP to AgrC2, or (iv) blocking the phosphorylation of AgrA2. Thus, quorum quenching can be used to control pathogenesis. To date, no QQ enzymes or QSIs active against *C. difficile* have been identified. Communication pathways of *S. aureus* share many similarities with those of *C. difficile* and may serve as a model in the search against anti-Agr compounds.

The first QQ molecule discovered in bacteria was AiiA, an enzyme from *Bacillus* species that inactivates acyl homoserine lactone (AH2), the QS signal of *Erwinia carotovora* (*Pectobacterium carotovorum*) [135]. Other examples of QQ enzymes are lactonase, acylase, oxidoreductase, and paraoxonase [136]. Ambuic acid, a fungal secondary metabolite, blocks the synthesis of AIPs in several Gram-positive bacteria, including *S. aureus* and *Listeria innocua* by inhibiting N-terminal cleavage by AgrB [137]. Apolipoprotein B sequestered QS signals produced by *S. aureus* [138]. Savirin, Naringenin, and Ω-hydroxy emodin (OHM) repress AgrA activity in *S. aureus* and inhibit the transcription of virulence factors [139]. Cochinmicin, avellanin, and solonamide B cyclodepsipeptide can also act as competitive inhibitors of the AgrC protein [140].

Anion-binding resins such as X-aptamers (small nucleic acid molecules) with high affinity to the N-terminal glucosyltransferase domain or the C-terminal receptor-binding domain of tcdA and tcdB may sequester or inactivate toxins [92]. Bile salts such as taurocholate, tolevamer, cholestyramines, and colestipol also have toxin-binding properties [92]. Tolevamer is a high-molecular-weight, soluble anionic polymer [141]. Darkoh et al. [92] reported that tolevamer, administered at 6 g per day, was as effective as vancomycin administered at 500 mg per day and was deemed effective in the treatment of mild to moderate *C. difficile* diarrhea. However, tolevamers are associated with an increased risk of hypokalemia. Nevertheless, patients treated with tolevamer reduced the severity of CDI [141] but were less efficient than vancomycin and metronidazole [110]. Cholestyramine and colestipol also attached to vancomycin and decreased its activity [141].

## 8. Immunotherapy

Immunization against *C. difficile* toxins, cell wall structures or outer-membrane proteins offers the prospect of a relatively low-cost approach to CDI prevention. Bezlotoxumab, approved by the FDA, was the first prophylactic antibody used to treat recurrent CDI [92]. The monoclonal antibody targeted TcdB. Actoxumab, a different antibody, targets the C-terminal receptor-binding domain of toxin A [142,143]. Actoxumab used in combination with bezlotoxumab prevented CDI when tested in mice and hamsters [142,144]. A prophylactic concentration of bezlotoxumab, alone or combined with actoxumab, provided 100% protection to piglets [145]. In humans, a combination of actoxumab and bezlotoxumab significantly reduced the rate of CDI relapse [146]. Several combinations of antibodies targeting toxins A and B are in the early stages of development and have been shown to be more effective than actoxumab and bezlotoxumab in the treatment and recurrence of CDI and diarrhea in hamsters [143].

Phase III clinical trials are underway to test multi-dose-inactivated *C. difficile* toxin-based (toxoid) vaccines. Sanofi Pasteur’s ACAM-CDIFF vaccine contains formalin-inactivated toxoid, and preliminary results from the Phase I trial demonstrated that it was safe and successful in eliciting an adequate neutralizing antibody response [121,147]. Pfizer is currently initiating a large-scale multicenter Phase III trial. Valneva et al. developed a vaccine using a recombinant fusion protein containing cell-binding domains from truncated forms of toxins A and B. It is not clear whether vaccination will be effective for primary or secondary prevention and whether it will prevent or reduce disease severity. Clinical utilization also depends on its efficacy, cost, and safety. Vaccination is unlikely to eliminate colonization; therefore, patient isolation is important for preventing CDI transmission.

Even though toxins A and B are the primary targets of most immunotherapies in development, several other virulence and colonization factors such as flagella, surface-layer proteins, Cpw84 proteins, and pilin are promising avenues for therapeutic intervention. Success in the development of drugs that target virulence and colonization factors may guide the next generation of CDI therapies. Currently, there are no therapies for CDI that are based on the direct inhibition of toxin production, toxin activity, or colonization factors. Bender et al. [148] are actively investigating novel approaches to CDI treatment. As antitoxin antibodies do not prevent *C. difficile* colonization [146], antibodies targeting cell wall proteins or adherent factors may be an answer. Previous studies have shown a significant decline in *C. difficile* colonization when mice were immunized with anti-flagellin (FliC) and flagellin filament cap protein (FliD) antibodies [149]. Hamsters orally administered purified FliD-specific antibodies were protected from CDI when challenged with *C. difficile* strain 630 [150]. Mice rectally vaccinated with FliD and cell wall extracts showed a significant decrease in *C. difficile* colonization [151]. Concluded from these studies, prevention of colonization may be the answer to the treatment of CDI.

## 9. Probiotics

Probiotics have been used effectively in the treatment of a variety of diseases, including CDIs, but their efficacy is strain- and disease-specific [152]. Probiotics may produce proteases that degrade TcdA and TcdB and compete with toxins for attachment to the gut wall [153]. *Lactobacillus acidophilus* GPIB, isolated from swine, reduced *C. difficile* virulence by decreasing AI-2 activity [154]. Downregulation of virulence genes was also observed. Heat-inactivated cell extracts of *L. fermentum* Lim2 suppressed *LuxS*, *tcdA,* and *tcdB* of *C. difficile* 027 [155]. Upregulation of the negative regulator gene (*tcdC*) was also recorded. Conclusive proof that probiotic strains interfere with the QS system of *C. difficile* by producing QQ enzymes or QSIs is lacking. The molecular structures and biochemical pathways of these inhibitors may provide insights into the control of CDI and the development of next-generation probiotics. Further research is needed on strains from the phyla Bacteroidetes and Firmicutes, as they have shown promising results. Probiotics have immunoprotective properties, hamper the adherence of C. difficile to the intestinal lumen, and modulate the host’s immune response. Ofosu [156] showed that probiotics increase the intestinal secretion of IgA antitoxin and inhibit the production of IL-8, a proinflammatory cytokine. *Lactobacillus* and *Bifidobacterium* strains are the most frequently utilized probiotics, in addition to the yeast *Saccharomyces boulardii* as adjuvant treatment in CDI or as primary prevention therapy for patients receiving vancomycin [121,157]. *S. boulardii* may, however, induce fungicemia in immunocompromised patients [121,158,159].

Fecal microbial transplants (FMTs) may be an option [160], as seen in clinical trials having shown more than 85% effectiveness [160,161]. Some evidence suggests that host secretions or microbial metabolites may also be sufficient for treatment of CDI [162,163]. FMT proved effective in preventing recurrences of CDI and treating refractory cases [164]. In one study, 91% of CDI patients who were refractory to antibiotics showed a positive response to FMT [160]. In another study, a cure rate of 75% with a single FMT infusion and 100% with multiple FMT infusions (administered to patients with severe CDI refractory to antibiotics) was reported [165].

A Phase II clinical trial of a non-toxigenic *C. difficile* strain M3 (NTCD M3) developed by Viropharma, Inc., demonstrated efficacy in reducing CDI recurrence, with possible restoration of the intestinal flora to its normal state [159]. It was reported that 22 weeks after administration, NTCD M3 strains could not be detected in stools. This observation suggests that colonization of NTCD M3 strains may be transient and presumably occurs because of the restoration of the normal microbiota, which may then provide protection against subsequent CDI [159].

## 10. Bacteriophage Therapy

Bacteriophages isolated from patients with CDI are non-lytic and belong to the Myoviridae (phiC2, phiC5, phiC8) and Siphoviridae (phiC6) subfamilies of Caudovirales [166]. Several *C. difficile* phages are, however, not behaving in a lysogenic manner and are thus not integrated into the host genome but remain episomal [167]. This is specifically the case with larger phages of approximately 130 kbp (i.e., phiCD5763, phiCD5774, phiCD211), although phiCD38-2 (41 kbp) also exists as a plasmid prophage [168]. Phage phiHN10 binds to S-layer proteins [169]. Interestingly, the cell wall protein CwpV from *C. difficile* confers resistance to infection by different phages, including members of the Siphoviridae and Myoviridae families [170]. The mode of activity is to prevent phage DNA from entering the cell [170]. Some phages, e.g., phiCDHM1 contain homologs of *agr* genes [171] that may promote the survival and replication of phiCDHM1 and its host. Phage phiSemix9P1 has a functional binary toxin locus (CdtLoc) [172], suggesting that lysogenic phages may play an important role in the spreading of toxin genes.

Prophylactic treatment with phage phiCD27 reduced *C. difficile* cell numbers significantly and prevented the production of TcdA and TcdB [173]. Treatment of *C. difficile* strains CD105LC2 and CD105HE1 with phages showed limited clearing of the cultures in vitro [174]. The most effective phage (phiCDHM2) killed almost all cells of *C. difficile* within 5 h, but growth recovered 24 h later [174]. A four-phage cocktail (phiCDHM1-phiCDHM2-phiCDHM5-phiCDHM6) proved effective in reducing CD105LC2 biofilms in vitro [175]. Despite reduced colonization, *C. difficile* was still detectable in the cecum and colon of most animals [174]. This indicated that phage cocktail treatment could delay, but not prevent, CDI.

The use of phage-derived endolysins and phage tail-like particles (PTLPs) against *C. difficile* has been explored in a few studies [176]. Several strains of *C. difficile* produce PTLPs when the SOS response is induced, resulting in the killing of competing strains [177]. For further background on PTLPs, the reader is referred to the reviews of Dams et al. [178] and Heuler et al. [179].

An endolysin targeting *C. difficile*, e.g., phiCD27-derived CD27L, was successfully overexpressed in *E. coli* and showed activity against all 30 strains, including two strains of the hypervirulent ribotype 027 [180]. Truncation of the endolysin to its N-terminal catalytic domain (CD27L1−179) enhanced lytic activity and broadened the host range [181]. The PlyCD catalytic domain (PlyCD1−174) also showed superior lytic activity. Combined with a vancomycin pre-treatment, PlyCD1−174 significantly reduced the titer of Cd in vitro [182].

Bacteriophage therapy may be a good alternative to the treatment of CDI, as they are highly species- or strain-specific and are therefore far less detrimental to the intestinal microbiota. Further research on endolysins needs to be conducted, as they display high species specificity and may be used in combination with antibiotic treatment. One of the challenges to overcome would be to safeguard phages and endolysins against the destruction by gastric enzymes.

## 11. Conclusions

The emergence of hypervirulent strains owing to increased antibiotic use is one of the reasons why CDI is considered a high-risk pathogen. Those that are most at risk are individuals suffering from IBD, immunodeficiency, and hypoalbuminemia, who underwent an organ transplant, had malignant tumors, and received chemotherapy. The prevention of biofilm formation and toxin production is probably the most promising alternative treatment for CDI. However, no therapeutic agents are available to inhibit colonization and toxin production or to suppress toxin activity. Antibodies may be the answer, but recipients of such treatments may generate secondary antibodies and run the risk of developing autoimmune diseases. Competitive exclusion of *C. difficile* and prevention of adhesion to receptors in the GIT is another option worth exploring but requires in-depth knowledge of changes in the gut microbiome of patients with CDI. Although the use of probiotics to relieve CDI has been reported, results have not been conclusive. The use of non-toxigenic *C. difficile* strains to outcompete toxigenic strains has also been proposed and is currently being evaluated in clinical trials.

## Figures and Tables

**Figure 1 microorganisms-11-02161-f001:**
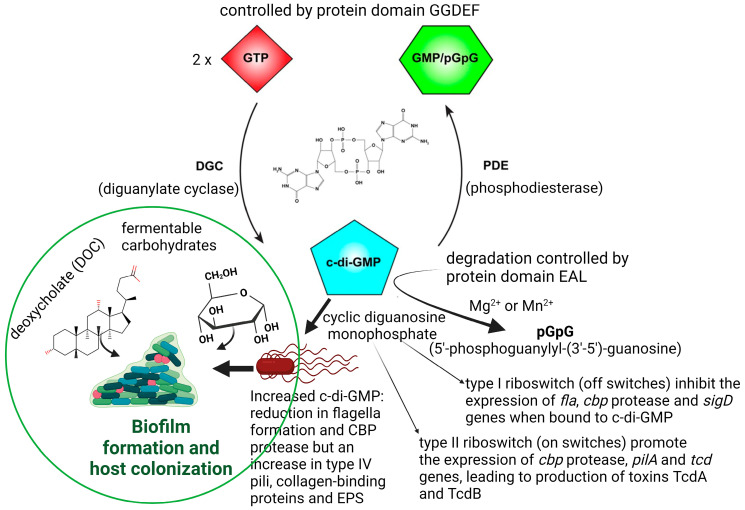
Cyclic di-GMP (c-di-GMP)-mediated riboswitches control the colonization of *C. difficile* to the host’s epithelial cells. Type I and II riboswitches control the expression of factors that are involved in motility, surface attachment, and virulence, including production of TcdA and TcdB. Type I riboswitches (off switches) inhibit translation following the binding of c-di-GMP, whereas type II riboswitches (on switches) promote the translation of target genes when bound to c-di-GMP. Increased levels of c-di-GMP stimulate the expression of adhesion factors, such as type IV pili and collagen-binding proteins (CBPs), and inhibit the expression of flagellar genes and CBP protease. Biofilm formation is altered by antimicrobials, QS signals, and the transcription factor spo0A that regulates sporulation. This illustration was made using BioRender (https://biorender.com/), accessed on 3 April 2023.

**Figure 2 microorganisms-11-02161-f002:**
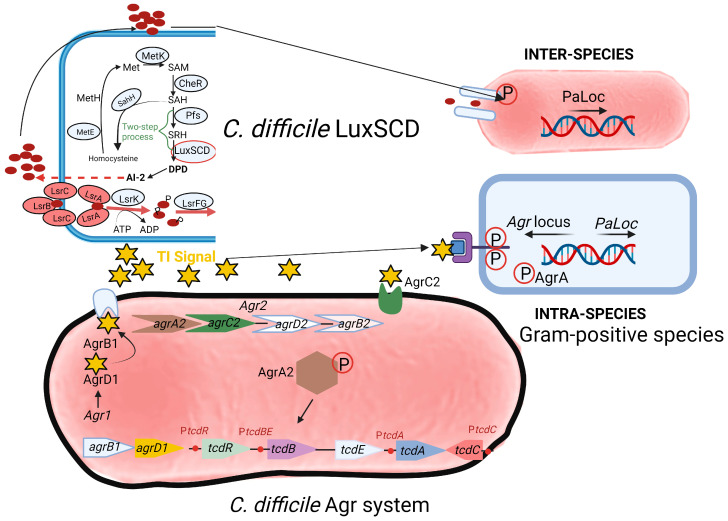
Quorum sensing (QS) in *C. difficile*. Two communication systems have been identified: an inter-species LuxSCD system (top section of diagram) and an intra-species accessory gene regulator (Agr) system (bottom section of diagram). The enzyme LuxSCD produces auto-inducer peptides (AI-2, red circles) that interact with receptors on another Gram-positive cell and is phosphorylated (top section of diagram). At a certain threshold, transcription of the *PaLoc* genes in the responding cell and the *LuxSCD* locus (not shown in responding cell) is activated. Intra-species communication (between two strains of *C. difficile*) occurs with TI (thiolactone peptide, depicted as yellow stars, bottom part of diagram). At a certain threshold, TI attaches to AgrC (a transmembrane protein, blue square in purple bracket), and AgrA is phosphorylated before it activates the transcription of the *Agr* and *PaLoc* loci. Two *Agr* loci (*Agr1* and *Agr2*) have been identified. The *Agr1* locus contains quorum signal generation genes *agrB1* and *agrD1* that encodes AgrB1 (a transmembrane protein) and AgrD1 (a prepeptide) responsible for production of TI. Sufficient levels of T1 may trigger TcdA and TcdB production. The *Agr2* locus contains quorum signal generation genes *agrB2* and *agrD2*, and response genes *agrC2* and *agrA2*. Hypervirulent strains, such as R20291, have loci *Agr1* and *Agr2*. Non-hypervirulent strains, such as 630, have only the *Agr1* locus. TI interacts with a two-component AgrC2 histidine kinase, which phosphorylates and dimerizes the response regulator AgrA2. This induces toxin production directly or indirectly. This illustration was made using BioRender (https://biorender.com/), accessed on 3 April 2023.

**Figure 3 microorganisms-11-02161-f003:**
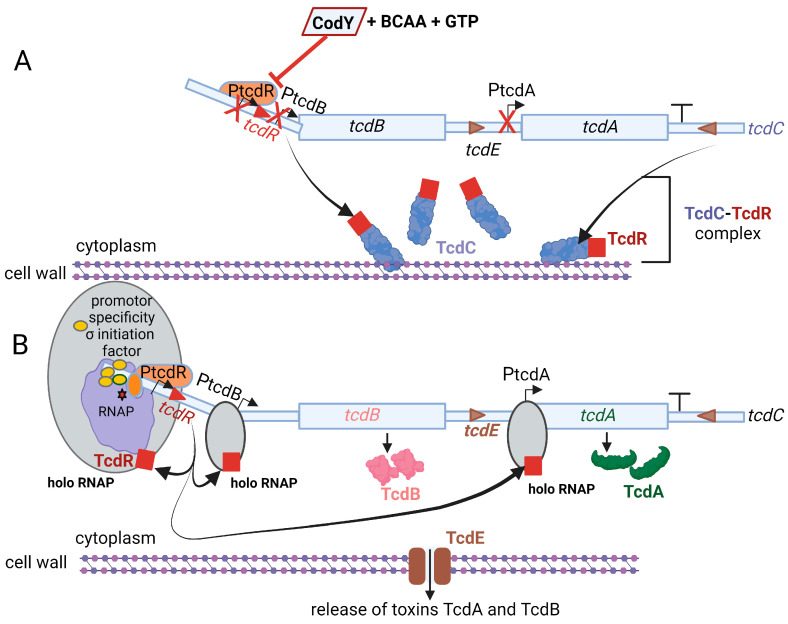
Pathogenicity locus (PaLoc) of *C. difficile* and regulation of toxin gene expression. For simplicity, genes in the PaLoc are depicted as light-blue rectangles. (**A**) During logarithmic growth, *tcdA*, *tcdB,* and *tcdR* (boxed) are expressed at low levels, regulated by a TcdR-independent promoter (PtcdR, orange). In the presence of GTP (guanosine triphosphate) and BCAAs (branched-chain amino acids, e.g., leucine, isoleucine, and valine), CodY (a transcriptional regulator of sporulation) binds to PtcdR and suppresses the expression of *tcdR*. This also leads to low expression of toxin genes *tcdA* and *tcdB*. The negative regulator gene, *tcdC* (on far right), is maximally expressed to produce the membrane-associated protein TcdC (in purple). The latter insulates minute amounts of TcdR (red squares) that may be expressed and further suppresses the expression of toxin genes. (**B**) During stationary (nutrient-limiting conditions or stress), the expression of toxin genes is induced. RNAP (depicted as a large purple circle), with an active site (red star), binds to a promoter-specificity σ initiation factor, also called the σ-specificity factor (yellow circle). The resulting holo RNAP (large gray circle on the left) reacts with promoters PtcdR, PtcdB, and PtcdA, respectively. The double-stranded DNA is loosely bound on the surface of RNAP. The two smaller gray circles also represent holo RNAP. In the absence of GTP and BCAA, CodY (not shown here) does not bind to PtcdR, and *tcdR* expression occurs unhindered. The *tcdC* gene (on far right) is not optimally expressed, which means TcdR is free to interact with RNAP and form the holozyme complex. The latter binds to the tcdR promoter (PtcdR) and promotors PtcdB and PtcdA, leading to maximal expression of *tcdA*, *tcdB*, and *tcdR* and thus high levels of toxins TcdA and TcdB. TcdE, a holin-like protein (depicted as brown rectangles in cell wall), facilitates the release of toxins TcdA and TcdB. This illustration was made using BioRender (https://biorender.com/), accessed on 3 April 2023.

**Figure 4 microorganisms-11-02161-f004:**
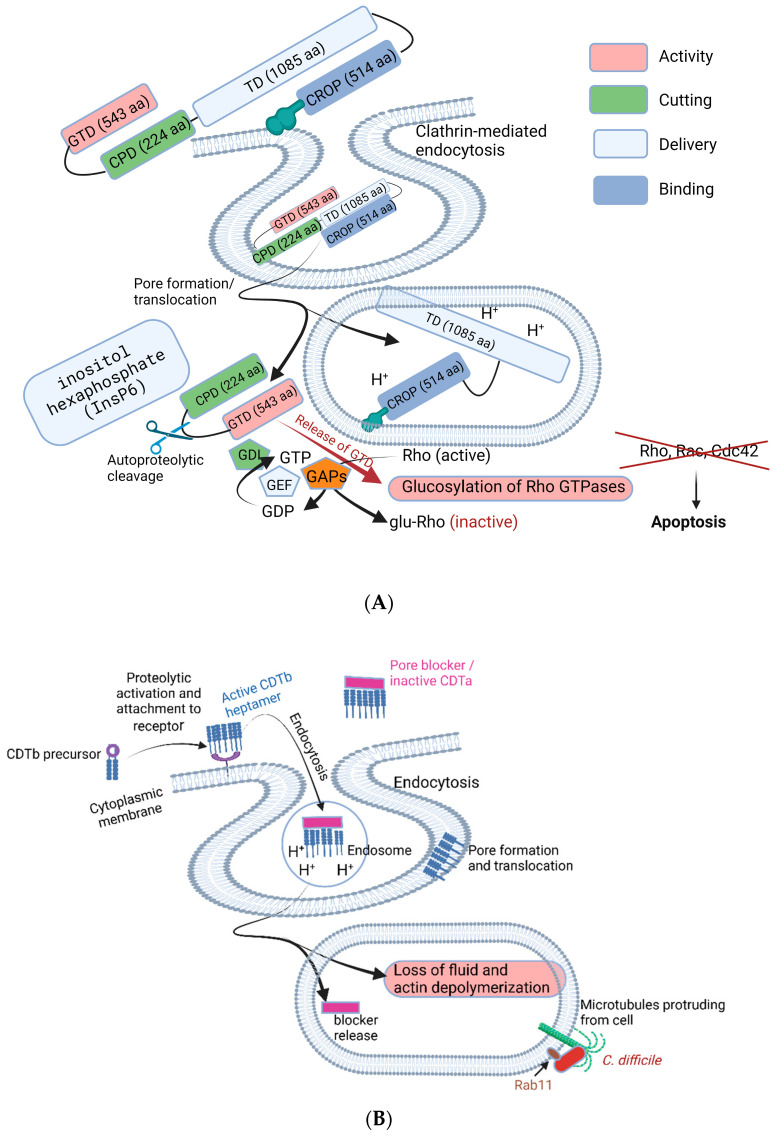
Mechanisms of *C. difficile* toxin pathogenicity. (**A**) The presentation is that of TcdB. Both toxins, TcdB and TcdA, bind to the surface of intestinal epithelial cells. The GTD (glucosyltransferase domain, in pink), located at the N terminus, is linked to CPD (cysteine protease domain, in green) involved in cleaving of the toxin. The TD (translocation domain, light blue) is involved in pore formation, conformational changes, and the delivery of the GTD and CPD. The CROP (combined repetitive oligopeptide, dark blue) binds to receptors on the cell membrane. After binding, the toxin is internalized, followed by endocytosis (top half of diagram **A**). The intracellular acidic environment of the endosome causes the toxin to undergo conformational changes and initiate pore formation (bottom part of diagram **A**). CPD and GTD are translocated across the endosomal membrane and then cleaved by a protease activated by inositol hexaphosphate (InsP6). GTD is released into the host cytosol and glucosylate Rho proteins to inactive glu-Rho. Activation and signal transduction of the Rho GTPases are regulated by a classic GTPase cycle. The cycle is controlled by regulatory proteins acting as GDIs (guanine nucleotide dissociation inhibitors, green triangle) that extract inactive Rho GTPase from membranes, GEFs (guanine nucleotide exchange factors, white triangle) that catalyze nucleotide exchange and mediate activation, and GAPs (GTPase-activating proteins, orange triangle) that stimulate GTP hydrolysis to GDP. (**B**) The active CDTb heptamer binds to the cell membrane, is enclosed in an endosome, and is then translocated across the membrane once the blocker (in pink) is released. This results in fluid loss and irreversible actin depolymerization, which leads to the formation of microtubule-based protrusions (depicted in green). Rab11 = Ras-associated binding proteins (small GTPases) of the Ras superfamily that cycle between the cytosol and different membranes. Not shown here are Rab5 and Rab4. This illustration was made using BioRender (https://biorender.com/), accessed on 9 May 2023.

**Figure 5 microorganisms-11-02161-f005:**
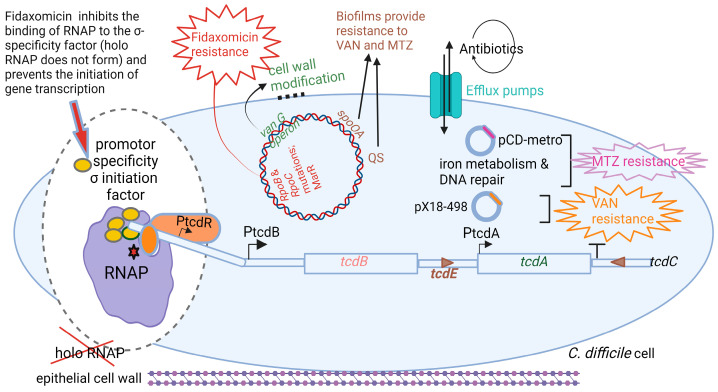
Resistance mechanisms *C. difficile* developed against metronidazole (MTZ), vancomycin (VAN), and fidaxomicin. Resistance to MTZ is encoded by plasmid pCD-metro and genes involved in iron and redox metabolism. VAN resistance is either encoded by plasmid pX18-498 or the vanG operon that modifies the peptidoglycan by replacing the terminal D-Ala with D-Ser. Mutations in *rpoB, rpoC,* and MarR homolog CD2212 lead to resistance to fidaxomicin. Cells in biofilm are less affected by MTZ and VAN. Deletion of the ATP-binding cassette transporter CD2068 in efflux pumps reduces the activity of MTZ 1.4-fold. This illustration was made using BioRender (https://biorender.com/), accessed on 11 May 2023.

## Data Availability

Not applicable.

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
