# Peer review of "Biofilm Formation of Clostridioides difficile, Toxin Production and Alternatives to Conventional Antibiotics in the Treatment of CDI"

_microorganisms, 2023, doi:10.3390/microorganisms11092161_

Round 1

Reviewer 1 Report

Review of ‘Where we stand with alternatives to antibiotics of Clostridioidis difficile infection’

The title of the manuscript is misleading as most of the paper is focused on C. difficile pathogenicity rather than on alternative treatments. I suggest you change the title of the paper to reflect the text.

The sections are very detailed and the text is very difficult to follow – I suggest the authors make the text more reader friendly. In each section the authors go into great detail without first introducing the strains. Such as in the section on ‘toxin production’ the first sentence is about accessory genes rather than stating C. difficile produces x, y and z toxin that contributes to their pathogenicity. The sections need to be re-written to make them more accessable.   

Abstract

The abstract is difficult to follow and needs to be re-written. It doesn’t state what the problem is, why alternatives to antibiotics are needed, what the aim of the review is and what the review will cover. With these points in mind the abstract needs to be re-written.

Introduction

·      Line 30 – ‘%’ needs to be written as percentage

·      Line 32 – which ‘authors’ are you referring to?

·      Again it’s not very clear why we need to look at alternative treatments, please state why and include this in the final paragraph on the introduction

·      Also need to include a paragraph on the structure of the paper and what details will be discussed

Section 2

·      Line 57 – the first sentence does not make sense, clarify

Quality of english is good.

Author Response

The title of the manuscript is misleading as most of the paper is focused on C. difficile pathogenicity rather than on alternative treatments. I suggest you change the title of the paper to reflect the text.

Answer: The title has been changed to “Biofilm Formation of Clostridioides difficile, Toxin Production and Alternatives to Conventional Antibiotics in the Treatment of CDI”

The sections are very detailed and the text is very difficult to follow – I suggest the authors make the text more reader friendly. In each section the authors go into great detail without first introducing the strains. Such as in the section on ‘toxin production’ the first sentence is about accessory genes rather than stating C. difficile produces x, y and z toxin that contributes to their pathogenicity. The sections need to be re-written to make them more accessable.

Answer: The text has been changed by adding introductory sentences where required (lines 209-212, 266-278, 524-531, 562 and 563).

Abstract

The abstract is difficult to follow and needs to be re-written. It doesn’t state what the problem is, why alternatives to antibiotics are needed, what the aim of the review is and what the review will cover. With these points in mind the abstract needs to be re-written.

Answer: The abstract (lines 9-31) has been rewritten.  The problem statement is mentioned in lines 9-13, the requirement for alternatives to antibiotics in lines 14-29, the aim of the review, and what the review covers in lines 29-31.

Introduction

Line 30 – ‘%’ needs to be written as percentage

Answer: changed to “One to three percent….”(line 43).

Line 32 – which ‘authors’ are you referring to?

 Answer: Reference to Mada and Alam [9] (now added in line 45).

Again it’s not very clear why we need to look at alternative treatments, please state why and include this in the final paragraph on the introduction

 Answer: Reasons given in lines 63-71.

Also need to include a paragraph on the structure of the paper and what details will be discussed

Answer: Included (lines 72-78).

Section 2

Line 57 – the first sentence does not make sense, clarify

Answer: The sentence has been changed to “CDI is contracted through the ingestion of endospores [18]”. (now line 80).

Reviewer 2 Report

This single author manuscript provides a comprehensive review of a somewhat-arbitrary topic -the “alternatives”  to antibiotic treatments of C difficile.   It includes a description of basic science pathways (eg Agr and quorum sensing)  and mechanisms  that could be targeted for future antibiotics, discussions on a newer antibiotic called fidaxomicin which somehow gets its own section(though not a “non-antibiotic”), bacteriophage therapy, immunotherapy and a very short section on probiotics mainly regarding their effects on quorum sensing. No mention is made of microbiome replacement therapies, which are the “hottest” topic in the field of C difficile therapeutics and beyond, and probably should be mentioned more prominently in a review with this title.   It seems the author wants the focus of the article to be the Agr and quorum sensing systems and how these may relate to CDI therapeutics – perhaps this focus may make more sense than the broad overview presented

Specific comments:

Abstract – This is a poor summary of the work, the weakest past of the work with an inordinate focus on Agr and quorum sensing and some factually troubling claims:

-Line 9 – the claim that treatment for CDI fails “as the cells form impregnable biofilms” is speculative as the formation of endospores may also protect the organism from antibiotic killing. .

Line 16 – regarding Monoclonal antibodies -“more human trials will have to be conducted to evaluate their efficacy” is not true -large scale studies have been completed comparable to other approved therapeutic products.  

Line 18 – “treatment of CDI with ..FMT produced fluctuating results” is also not fair – a vast majority of studies show exceptionally good results -and this data is barely mentioned in this work.

General – why mention loss of motility (minimally discussed), QQ molecules that have not been identified, Fidaxomycin which does not qualify as an antibiotic alternative, tolevamer which is not being produced, and probiotics only vis-à-vis their  potential effects on the QS system.

Introduction

Line 26 – “severe”  is not correct -any exposure to antibiotics can precipitate CDI. Also, the period should be a comma.

Section 3

Lines226-229 seem to be contradictory – indicating that a LuxS mutant could not form biofilms, and then in the next line talking about RNA sequencing of the luxS mutant grown in biofilm. Based on earlier text, the authors should clarify whether/that the luxS mutant was grown in DPD- supplemented media to allow biofilm formation.

Section 4 –Toxin production

Should reorganize this to start with a discussion of the toxins including Fig 3, before launching into explanations of how agr expression affects toxin production. Section 5 could be included in the introductory paragraphs  of this section as well.

Lines 302-304 do not belong in the toxin section

Section 6

Line426-27- “half of the antibiotics used to treat CDI are ineffective ” needs a reference

Lines 427-34 do not relate to the subject topic –which is treatment of CDI , but rather relate to antibiotics as  risk factors for CDI.

Lines 435-436. -”General practice’ references an article from2016- current practice guidelines (ref 134) no longer advocate use of metronidazole.

Line 451 – resistance to MNZ and VAN is very rare- more accurate is to cite “failure” of treatment with MNZ and VAN –which was largely NOT due to identifiable  resistance

Section 7 –fidaxomicin section- should be collapsed into the prior section. It should not be a focus of this review but rather part of the background. Figure 5 can be removed.

Section 8 –should be retitled “toxin binding or suppression” – as lines 534-544 relate to toxin binding resins, not  repressed toxin production. Lines 545-52 form the introduction to this section and should be first.

Section 9 should be labelled “immunotherapy” rather than immunization.

Lines 561-64 – should note in the text of lines 561-564 that reference 155 relates to hamster studies -

Section 10 should be labelled “probiotics” - as this section only peripherally deals with QS.

Lines 613-614 mention facal microbial transplants –besides correcting the spelling of fecal, this topic deserves more than 1 cursory mention with a reference to a study of FMT in severe disease, which is a less common use of this modality. It has been more extensively used in preventing relapse in patients with frequent recurrences of disease, with ample reviews indicating superior efficacy in this difficult to cure population of patients.

Lines 675-678 more properly belong at the end of section11.

only minor issues noted

Author Response

No mention is made of microbiome replacement therapies, which are the “hottest” topic in the field of C difficile therapeutics and beyond, and probably should be mentioned more prominently in a review with this title. It seems the author wants the focus of the article to be the Agr and quorum sensing systems and how these may relate to CDI therapeutics – perhaps this focus may make more sense than the broad overview presented.

Answer: A section on fecal microbial transplants has been added and eight papers have been referenced (lines 622-629).

Specific comments:

Abstract – This is a poor summary of the work, the weakest past of the work with an inordinate focus on Agr and quorum sensing and some factually troubling claims:

-Line 9 – the claim that treatment for CDI fails “as the cells form impregnable biofilms” is speculative as the formation of endospores may also protect the organism from antibiotic killing.

Answer: The abstract has been rewritten, as also requested by one of the other reviewers.  The sentence referred to has been changed to “…..poor ability to penetrate biofilms and the formation of endospores.” (line 16).

Line 16 – regarding Monoclonal antibodies -“more human trials will have to be conducted to evaluate their efficacy” is not true -large scale studies have been completed comparable to other approved therapeutic products.

Answer: Although large-scale studies have been conducted, my opinion is that more human trials are still required and the sentence has been changed to “      more human trials are still required to evaluate their efficacy.” (line 24).

Line 18 – “treatment of CDI with FMT produced fluctuating results” is also not fair – a vast majority of studies show exceptionally good results -and this data is barely mentioned in this work.

Answer: The sentence has been changed to “Treatment of CDI with fecal microbial transplants (FMTs), on the other hand, produced promising results.” (lines 28 and 29).  A section on fecal microbial transplants has been added to the text and eight papers have been referenced (lines 622-629).

General – why mention loss of motility (minimally discussed), QQ molecules that have not been identified, Fidaxomycin which does not qualify as an antibiotic alternative, tolevamer which is not being produced, and probiotics only vis-à-vis their potential effects on the QS system.

Answer: These topics were addressed to give the reader a broader overview of the subject and point out areas that require further research on CDI treatment.

Introduction

Line 26 – “severe”is not correct -any exposure to antibiotics can precipitate CDI. Also, the period should be a comma.

Answer: This is a valid point and was corrected with the amendment of the abstract.

Section 3

Lines226-229 seem to be contradictory – indicating that a LuxS mutant could not form biofilms, and then in the next line talking about RNA sequencing of the luxS mutant grown in biofilm. Based on earlier text, the authors should clarify whether/that the luxS mutant was grown in DPD- supplemented media to allow biofilm formation.

Answer: The RNA sequencing refers to the other genes mentioned and not luxS, i.e. expression of “CDR20291_2554 (crr), a phosphotransferase (PTS) glucose-specific transporter subunit IIA; CDR20291_2927, a cellobiose phosphate-degrading protein; and CDR20291_2930 (treA), a trehalose-6-phosphate hydrolase” (lines 253-257).

Section 4 –Toxin production

Should reorganize this to start with a discussion of the toxins including Fig 3, before launching into explanations of how agr expression affects toxin production. Section 5 could be included in the introductory paragraphs of this section as well.

Answer: Valid point. Section 4 has been amended by first introducing the toxins (lines 266-278).  Section 5 was left as a section on its own, as this addresses the mode of action of the toxins, with reference to Figure 4.

Lines 302-304 do not belong in the toxin section.

Answer: The sentences “Antibiotics currently used to treat CDI (Fidaxomicin, MNZ and VAN) are active against vegetative C. difficile cells, but not endospores.  Fidaxomicin did not significantly reduce recurrence in patients infected with hypervirulent strain B1/NAP1/027 [86].” have been deleted (see end of line 341).

Section 6

Line426-27- “half of the antibiotics used to treat CDI are ineffective ” needs a reference

Answer: The sentence has been deleted (see line 450).

Lines 427-34 do not relate to the subject topic –which is treatment of CDI , but rather relate to antibiotics as risk factors for CDI.

Answer: This section has been deleted from the first paragraph in section 6 (starting at line 449) and excluded from the review.

Lines 435-436. -”General practice’ references an article from2016- current practice guidelines (ref 134) no longer advocate use of metronidazole.

Answer: The sentence “The general practice is to treat mild cases of CDI with oral administration of MNZ and VAN for 10 to 14 days [114].” has been deleted from section 6.

Line 451 – resistance to MNZ and VAN is very rare- more accurate is to cite “failure” of treatment with MNZ and VAN –which was largely NOT due to identifiable resistance

Answer: The sentence has been changed to “Failure of treatment with MNZ and VAN led to the treatment of CDI with Fidaxomicin [114,115].” (line 465).

Section 7 –fidaxomicin section- should be collapsed into the prior section. It should not be a focus of this review but rather part of the background. Figure 5 can be removed.

Answer: Information in section 7 is now included in section 6 and Figure 5 has been deleted. The Figures were renumbered and the Figure numbers in the text were corrected.

Section 8 –should be retitled “toxin binding or suppression” – as lines 534-544 relate to toxin binding resins, not  repressed toxin production. Lines 545-52 form the introduction to this section and should be first.

Answer: The heading of section 8 was changed to “Toxin Binding or Suppression” (line 523).  The section has been changed by moving the paragraph up (see highlighted section, lines 524-531).

Section 9 should be labelled “immunotherapy” rather than immunization.

Answer: Section 9 was renamed to Immunotherapy (line 561).

Lines 561-64 – should note in the text of lines 561-564 that reference 155 relates to hamster studies –

Answer: Corrected (line 573).

Section 10 should be labelled “probiotics” - as this section only peripherally deals with QS.

Answer: Section 10 is now renamed to Probiotics (line 601).

Lines 613-614 mention facal microbial transplants –besides correcting the spelling of fecal, this topic deserves more than 1 cursory mention with a reference to a study of FMT in severe disease, which is a less common use of this modality. It has been more extensively used in preventing relapse in patients with frequent recurrences of disease, with ample reviews indicating superior efficacy in this difficult to cure population of patients.

Answer: The spelling was corrected to fecal and a section on fecal microbial transplants was added (lines 622-629). Papers on the topic were cited and the numbering of references was corrected throughout the paper.

Lines 675-678 more properly belong at the end of section11.

Answer: The paragraph was removed to the end of section 11 (lines 673-678).

Reviewer 3 Report

This is a comprehensive work on the intracellular mechanisms related to the physiology, biofilm formation, and toxin production by C. difficile. Provides a large amount of information that is more useful for basic researchers than for clinicians. May support and inspire research studies related to the prevention of infection and treatment of this bacterium.

Specifics comments:

Please change “Clostridiodis” to “Clostridioides”

Bezlotoxumab, actoxumab, tolevamer and the name of the antibiotics should be lowercase.

Line 26. Please remove "severe".

Line 30. Please remove the % sign.

Author Response

This is a comprehensive work on the intracellular mechanisms related to the physiology, biofilm formation, and toxin production by C. difficile. Provides a large amount of information that is more useful for basic researchers than for clinicians. May support and inspire research studies related to the prevention of infection and treatment of this bacterium.

Specifics comments:

Please change “Clostridiodis” to “Clostridioides”

Answer: Now changed throughout the paper (e.g. line 3, 9, 32, 37 and 701).

Bezlotoxumab, actoxumab, tolevamer and the name of the antibiotics should be lowercase.

Answer: All corrected (lines 23, 563, 568 and 570).

Line 26. Please remove "severe".

Answer: Now removed (line 39).

Line 30. Please remove the % sign.

Answer: Now removed (line 43).

Round 2

Reviewer 1 Report

The author has improved the manuscript and taken on board our previous comments. I suggest some minor changes below

Abstract

The abstract in chunky and difficult to follow – I suggest the authors has a slow read of it. I suggest the authors state these are the potential treatments that can be used to treat CDI and then list them instead of describing them individually.

Author Response

16 Aug 2023

Dear Editor

The abstract has been amended as suggested by the reviewer. Changes are highlighted.

Yours sincerely

Prof LMT Dicks
